# A Mixed Methods Approach to Culture-Sensitive Academic Self-Concept Research

**Lilith Rüschenpöhler *** and **Silvija Markic**

Department of Chemistry and Chemistry Education, Ludwigsburg University of Education,
71634 Ludwigsburg, Germany; markic@ph-ludwigsburg.de
* Correspondence: rueschenpoehler@ph-ludwigsburg.de; Tel.: +49-7141-140-766

**Abstract:** (1) Background: In self-concept research, Likert scales are still relied upon despite the fact that they pose methodological difficulties for research in culturally diverse societies. This calls the validity of the data into question. In the present study, we develop a mixed methods design for culture-sensitive academic self-concept research. We test it in a study about chemistry self-concept with secondary school students; (2) Methods: Interview ($N = 43$) and questionnaire ($N = 116$) data were collected; (3) Results: The mixed methods approach allowed connecting self-concept with culturally shaped narratives: in the quantitative data, we found the well-documented gender gap in favor of boys. However, among the students with a Turkish migration background, the girls showed stronger chemistry self-concepts. The interviews suggested that girls with Turkish migration background find it easier to connect their chemistry learning to their personal life than the boys with Turkish migration background; (4) Conclusion: Based on further literature, we hypothesize that these differences might be due to a less masculine conception of science in the Turkish society. The mixed methods approach allows detecting measurement bias, which increases the validity of science self-concept data in culturally diverse contexts.

**Keywords:** self-concept; gender; cultural background; mixed methods; chemistry

## 1. Introduction

Science self-concept research faces severe measurement difficulties when conducted in culturally diverse groups. A systematic literature review [1] showed that most research uses self-report data from Likert-type items. However, this produces difficulties in the measurement process because we do not know if the items have the same meaning in culturally different groups of students [2]. This problem persists in self-concept research since we still lack theoretical and methodological approaches for investigations of self-concept in culturally diverse societies [3].

In this article, we propose one possible solution to the prevailing methodological difficulties in academic self-concept research. We develop and test an approach for culture-sensitive academic self-concept research. Instead of relying exclusively on quantitative data, we suggest that a mixed methods approach might enhance the validity of the data. This is innovative in academic self-concept research since research in this area relies almost entirely on quantitative data [1] despite methodological difficulties. This approach can be employed in academic self-concept research irrespective of the subject that is studied. In this article, we will focus on chemistry since we dispose of expertise in this field. For assessing the use of the approach, we tested it in a study on chemistry self-concepts of secondary school students in the North of Germany. The investigation of chemistry self-concepts with the developed mixed methods approach serves to explore potentially fruitful paths for further research on self-concept in culturally diverse societies. In the present study, the mixed methods approach allows

(i) evaluating potential measurement bias and (ii) linking the self-concept-data to the associations of science with masculinity that might differ between cultures.

## 2. Theoretical Background

### 2.1. Methodological Difficulties in Self-Concept Research

Important measurement difficulties remain unresolved in self-concept research. When used in cross-cultural research or in culturally diverse societies, the established methods in self-concept research face methodological difficulties. This was shown by Barbara Byrne [3] who identified "a grave need for researchers to move beyond the paper-and-pencil approach to self-report measurement" (p. 904) because responses to self-concept questionnaires will be "influenced by a cultural bias that ultimately leads to differential perceptions of self" (p. 903). Byrne and colleagues [2] further categorize this error into two types: conceptual non-equivalence (self-concept means different things in different cultures) and measurement non-equivalence (bias caused by the instrument, e.g., because of culturally different response styles). Byrne [3] suggested using interviews alongside the established self-concept measures in order to control for this type of bias. However, the use of qualitative data in science self-concept research remains extremely rare, as we have pointed out in a systematic analysis of current self-concept research [1].

### 2.2. Development of an Approach to Culture-Sensitive Academic Self-Concept Research

Based on these considerations, the question remains how academic self-concept could be investigated in a more culture-sensitive way. A mixed methods design would be promising, following Byrne's [3] call for the combination of qualitative and quantitative data in self-concept research. Since investigations based on the prevailing theoretical models of self-concept are almost exclusively quantitative in nature [1], critical scrutiny of the theoretical foundations of science self-concept research is needed.

The prevailing models of self-concept allow assessing the strength of self-concepts. They therefore lay the foundation of quantitative assessments of science self-concept. The most prominent conception is the Shavelson model [4] which is, despite the fact that it was developed in the 1970s, still one of the key references in self-concept research. The authors define self-concept as "a person's perception of himself" (p. 411) and suggest that people have different self-concepts for different aspects of life that vary in strength. Science self-concepts belong to the academic domain and constitute a person's evaluation of his or her abilities in science. Building on this foundation, Herbert W. Marsh [5] developed the internal and external frame of reference model for research on academic self-concept which is still widely used.

However, for exploring qualitative differences in self-concept, the prevailing models provide little basis. A different approach is needed. For the purpose of this study, we use John Hattie's [6] model of the self for exploring qualitative dimensions of self-concept. The model focuses on psychological processes at the micro level. Hattie [6] emphasizes the active role of the self in the production of self-concepts. He does so by conceptualizing the self as a chooser. This allows investigating why a person adopts a specific self-concept. For example, a student's negative self-concept in science would be explained on the level of the following self-processes [6]:

1.  *Protecting the self.* A student can choose a negative self-concept in science in order to protect his self, for example, because he does not achieve as well in science as his classmates. The acceptance of his weak abilities could lower the pressure to keep up with his classmates. This can protect the self.
2.  *Preserving the self.* People seek long-term stability, i.e., the preservation of the self. If a student in the past achieved low in science and got good grades only recently, the student might still stick to a negative self-concept. Changing the self-concept could endanger the student's identity as a non-science person. Preserving his negative self-concept can, therefore, be a rational choice.

3.  *Promoting the self.* If a student identifies as a language person and couples this with a distinction from the natural sciences, adopting a negative science self-concept can strengthen the student's identity as a language person. This can promote the self.

Hattie's model shifts the focus on processes. Self-concept can, therefore, be studied with a focus on the individual rationales underlying self-concept. According to Hattie [6], a wide range of psychological processes are involved in the construction of the self. They all serve to protect, to promote, or to preserve the self.

In the academic domain, learning goal orientations (LGO) and performance goal orientations (PGO) are of special importance [6]. For the purpose of the present study, we concentrate on these in order to gain first insights into the construction of students' science self-concepts. In the following, we will briefly describe learning and performance goal orientations, mainly based on the fundamental works of Dweck and colleagues which still constitute major building blocks in current literature on learning and performance goal orientations.

Learning and performance goal orientations can be thought of as two different mindsets. When people activate their learning goal orientation (LGO), they tend to interpret situations as opportunities to learn. This implies that they perceive themselves as capable to learn. This is called an incrementalist theory of intelligence [7] which relies on the belief that they can control the learning process [8]. In practice, these students tend to take more risks in academic situations (e.g., they give an answer in class even if they are not sure if it is correct) because these situations bear the potential for learning and personal growth. This chance appears to be of greater importance than the potential failure [9]. It makes it easier for them to accept failure because they tend to focus on the learning effort they made rather than on success [7]. This leads to persistence when confronted with difficult tasks [9].

In contrast, when people activate their performance goal orientation (PGO), they tend to interpret situations as contexts in which they need to demonstrate their abilities [10]. This leads to important differences. When students adopt a PGO, they tend to believe that people have certain abilities but they can only improve them to a minor extent. They perceive low personal control over the development of their competences. The locus of control [8] appears to be external because their abilities appear as predefined and fixed which is called an entity theory of intelligence [7]. This motivates students to present their abilities rather than to strive to develop them [11]. With a performance goal orientation mistakes are difficult to deal with because they can appear as a personal failure [12]. Students who are performance goal-oriented tend to give up more easily and to encounter feelings of helplessness because they assume their abilities to be fixed [9]. Failure tends to be explained with personal incompetence [12], which can lead to obtrusive thoughts disrupting the work process [11]. Social goals and the desire to belong to the group become more important and students are more likely to give up the task at hand [11].

Based on these considerations, academic self-concept research could become more culture-sensitive through the adoption of a mixed methods approach. The Shavelson, Hubner, and Stanton model [4] provides the basis for investigating the strength of self-concept. The qualitative dimension of self-concepts can be explored using Hattie's [6] model.

### 2.3. The Relationship between Science Self-Concept and Culture

The study presented in this article tests the mixed methods approach for more culture-sensitive academic self-concept research using a concrete example, namely chemistry self-concept. In the following, we will summarize some key findings about the relation between science self-concept and culture, as well as findings about chemistry self-concept.

Despite the prevailing methodological difficulties, cultural differences in science self-concept have received considerable attention in science education research. The gender relations in science self-concept are a well-researched area. Since gender is a cultural construct [13], the gender relations touch differences in science self-concept that are associated with cultural categories. In numerous

studies, it has been shown that boys tend to have stronger science self-concepts in Western societies than girls. This has, for instance, been shown for the U.S. [14], Germany [15], and the U.K. [16]. However, this gender gap might depend on the cultural context since it was not found in a study conducted in Malaysia and Singapore [17] and in some other cases. One hypothesis could explain these findings: in some countries, science might not be associated as strongly with masculinity as it tends to be in Western countries (Figure 1, left). However, the present literature does not yet allow for a definite answer.

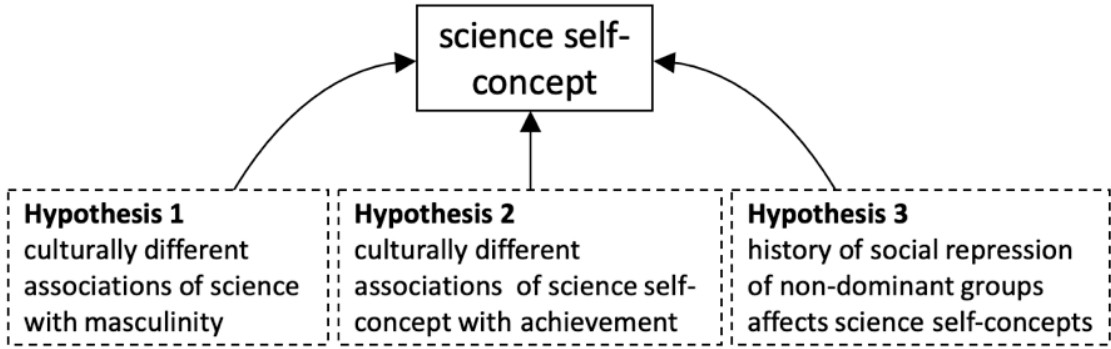

**Figure 1.** Hypotheses regarding possible influences of culture on science self-concept derived from literature by the authors.

In addition, the relation between science self-concept and achievement has been of central interest in investigations of the impact of culture on science self-concept. While the relation is positive in general [18], research indicates that it differs between cultures. In Asian and Eastern European countries, science self-concept levels are relatively low compared to the relatively high achievement levels in these regions [19]. This comes to a surprise because one would expect to find high levels of science self-concept in countries where the students achieve well. However, this finding suggests that self-concepts might play a different role for students in Asian and Eastern European countries than it does for students in Western countries (Figure 1, center). The reason for this difference seems to remain an open question.

Besides the differences in self-concept that seem to exist between countries, some studies identified effects of the cultural backgrounds of students living in the same country. In the U.S., students belonging to non-dominant ethnic groups tend to have lower self-concepts in science than students belonging to the dominant ethnic group [14]. This seems to be a pattern in some other Western countries as well: the same relation was found in Australia and New Zealand [20] and in the U.K. [21]. However, the study conducted in the U.K. [21] showed that students with an East Asian background are an exception. They show particularly strong self-concepts in science when living in Western countries.

The reasons for these differences between students with various cultural backgrounds living in the same country remain only partly understood. One factor explaining lower science self-concepts in students from a non-dominant ethnic group could be their socio-economic status which tends to be lower in many ethnic groups that have experienced or still experience repression, as Amanda Woods-McConney and colleagues [20] hypothesize (Figure 1, right). The students' socio-economic status is indeed strongly related to science self-concept [22]. Following this line of argumentation, it would be important to take a more holistic view in self-concept research, considering racial and social conflicts that existed in the past or that still exist. Australia, New Zealand, the U.K., and the U.S. all have a history shaped by colonialism and/or repressive policies that marginalized certain ethnic groups living in the country. The children from these groups now tend to show weaker science self-concepts. We believe that these phenomena can only be understood if the respective histories and present racial conflicts are considered.

These selected findings give an impression of the close intertwinement of science self-concept with culture. The associations of gender with science self-concept seem to differ between cultures, and likewise does the association with achievement. Further, the cultural backgrounds of students living in the same country seem to shape their science self-concepts. These findings suggest that students' thoughts about their abilities in science are embedded in the cultural context.

Chemistry self-concept tends to be less well understood than science self-concept. For research with secondary school students, no thoroughly tested and validated measurement instrument seems to be available. The existing well-tested instruments are designed for young adults in higher education, such as the Chemistry Self-Concept Inventory (CSCI) [23] and the Attitude toward the Subject of Chemistry Inventory (ASCI) [24] (revised version: [25]). These scales are based on Marsh's Self-Description Questionnaire III (SDQ) [26] which allows assessing the self-concepts of young adults. Studies tend to concentrate on chemistry self-concept in higher education, e.g., on the assessment of chemistry self-concept in a general chemistry course [27]. In addition, we are currently unaware of studies investigating the relation of chemistry self-concept with students' cultural backgrounds.

## 3. Research Goal and Questions

The goal of the present study is to explore the approach for a more culture-sensitive academic self-concept research developed in Section 2.2. We aim at developing an approach for culture-sensitive academic self-concept research for all subjects. In order to be culture-sensitive, a mixed methods design is needed [3]. We want to know:

*TQ (theoretical question).* To what extent can a mixed methods approach provide a basis for culture-sensitive academic self-concept research?

We intend to answer this question through the application to a concrete study. In this study, we concentrated on one concrete example of academic self-concept research, namely chemistry self-concept. We conducted an explorative study in which we investigated secondary school students' chemistry self-concepts. Special attention is turned towards the influences of gender and the students' cultural backgrounds on chemistry self-concept because these seem to influence science self-concepts (Figure 1). In this study, we want to assess both the strength of chemistry self-concept and its qualitative dimensions:

*EQ1 (empirical question 1).* How strong are the chemistry self-concepts of students with different cultural backgrounds and gender living in Germany?
*EQ2 (empirical question 2).* In what ways are the students' chemistry self-concepts associated with learning and performance goal orientations?

## 4. Materials and Methods

### 4.1. Type of Mixed Methods Design

We chose a concurrent triangulation mixed methods design [28] with questionnaires and interviews, based on both the perspective of the Shavelson, Hubner, and Stanton [4] model and the perspective of Hattie [6]. According to the criteria defined by John W. Creswell et al. [28], our design can be classified as a concurrent triangulation design: (i) the implementation was concurrent, i.e., the collection of qualitative and quantitative data took place at the same point in time. This design was chosen because we sought to compare the two data types in order to understand their interrelationships; (ii) Both data types were of equal priority; (iii) The integration of the two data types took place at data interpretation, i.e., data collection and data analysis were conducted separately for the data.

### 4.2. Instruments

*Questionnaires.* For assessing the students' chemistry self-concepts following the Shavelson model [4], we employed a chemistry self-concept scale developed by the authors (Table 1). The questionnaire was composed of nine Likert-type items covering both general aspects (items 1, 2, 3, 5) and more specific aspects such as the students' perception of their abilities in specific situations (items 3, 4, 6, 7), and life out of school (items 8, 9). The scale was piloted prior to the study and showed good reliability with values for Cronbach's α of 0.86. The students' cultural backgrounds were conceptualized with reference to the concept of migration background which accounts for the migration processes that took place in Germany after World War II. As defined in the official 2013 census [29], a person who was born in another country or whose parents have been born in another country has a migration background. The type of migration background was further specified by naming the respective countries.

**Table 1.** Mean values and standard deviations of the translated version of the chemistry self-concept scale ($N$ = 116); the original in German language can be requested from the authors.

| Item | M (SD) |
|---|---|
| 1. Chemistry is one of my strengths. | 3.48 (0.96) |
| 2. In chemistry, I'm one of the better students. | 3.35 (1.11) |
| 3. I am good at doing experiments. | 4.04 (0.78) |
| 4. When I have done an experiment, I understand what the result means. | 3.97 (0.83) |
| 5. I usually get on well in chemistry classes. | 3.98 (0.96) |
| 6. I understand the texts we read in chemistry. | 3.85 (0.87) |
| 7. I'm good at solving mathematical problems in chemistry. | 3.45 (1.27) |
| 8. Sometimes my chemistry knowledge helps me understand things in everyday life. | 3.38 (1.09) |
| 9. I would be able to choose a profession in which chemistry knowledge is needed. | 2.86 (1.23) |

*Interviews.* In addition, we conducted semi-structured interviews with some of the students who completed the questionnaire. Here, we investigated the dynamics of chemistry self-concept through the lens of Hattie's model [6]. In line with Hattie [6], we focused on the students' learning and performance goal orientations as two important self-processes. The interviews were divided into three parts. (i) *Warm-up.* At the beginning of the interviews, we asked the students to share some of their experiences from chemistry class. In particular, we wanted to know what they like and what they dislike about it; (ii) *Handling success and failure.* We wanted to know how they deal with success and failure in chemistry class because this behavior is an indicator of learning goal orientation (LGO) and performance goal orientation (PGO) [12]. We asked them to describe a situation in which they had been satisfied with themselves and one situation in which they had been dissatisfied with themselves in chemistry. These attributions can indicate LGO and PGO; (iii) *Dealing with difficult tasks.* We asked them how they deal with difficult tasks. Here, we were interested in their attitude towards situations in which, on the one hand, they face obstacles but which, on the other hand, also open the opportunity to engage in deep learning processes. This can reflect LGO and PGO as well [9].

### 4.3. Sample

The study was conducted in accordance with the Declaration of Helsinki [30], and the protocol was approved by the Ethics Committee of the state of Bremen (local ministry). We informed the teachers, the parents, and the students in a letter about the aims of the study, the procedure, and the voluntary nature of the study, and obtained written consent. We emphasized that non-participation did not have any negative consequences and that students could withdraw their answers at any moment. These instructions were repeated orally before conducting the study in class. In order to protect the students' anonymity in the study, a non-traceable code was produced that allowed to match the interviews with the corresponding questionnaires.

*Questionnaire data.* We collected questionnaire data from 116 students in the course of three months. In the part of Germany where data collection took place, it depends largely on the individual schools and teachers when chemistry topics are introduced in class. We, therefore, asked the teachers prior to the data collection process, if they had already taught chemistry in the respective group. The students of our sample were aged 11–19 ($M = 15.2$, $SD = 1.4$) and attended six different public secondary schools. The large majority of students fell in the age range of 13–17 years which is typical for these grades. Those who are younger usually skipped a grade. The students who are older usually either had to repeat a school year because they did not achieve well, or they are immigrants who first attended a separate language class before entering the regular school system. With this age range and the different school contexts, we intended to gain insights into experiences with chemistry across different contexts. In total, 54.3% of the students who filled in the questionnaire were female, and the majority of the students reported to have a migration background (54.3%). The concept of migration background classifies students along national lines and does not put them into larger categories such as 'Asian' or 'Black' as the U.S. American concept of race does. Nevertheless, we tried to group the students. The largest group was composed of students with a Turkish and/or Kurdish background (Table 2, 23.3%; $N = 27$). We assembled the students with a Turkish and a Kurdish background in one group because all students with a Kurdish background also reported having a Turkish migration background. As a simplification, we will refer to these as students with a Turkish migration background.

**Table 2.** Migration backgrounds of the students in the sample.

| | *N* | |
|---|---|---|
| | **Questionnaire** | **Interviews** |
| None | 53 (45.7%) | 19 (44.2%) |
| Turkish | 27 (23.3%) | 9 (20.9%) |
| Eastern European/Russian | 9 (7.8%) | 6 (14.0%) |
| South Asian | 8 (6.9%) | 3 (7.0%) |
| Sub-Saharan African | 7 (6.0%) | 3 (7.0%) |
| Others | 12 (10.3%) | 3 (7.0%) |
| Total | 116 (100%) | 43 (100%) |

*Interview data.* Forty-three of the students from the quantitative study volunteered to be interviewed after filling in the questionnaire (Table 2). The students were interviewed in pairs because we intended to make them feel at ease during the interviews, thereby increasing the content validity of the data. We, therefore, asked them to pick a classmate who would like to be interviewed as well. However, some of the students were interviewed alone because they did not find a partner. The interviews lasted about 10–15 min. In the interview study, 60.5% ($N = 29$) of the participants were female. Students with a migration background made up for 55.8% ($N = 24$). The students with a Turkish background constituted the largest group with 20.9% ($N = 9$) among the students with a migration background. The interviews were conducted in German language. The extracts quoted in this article have been translated by the authors.

*4.4. Data Analysis*

*Quantitative analyses.* First, we calculated Cronbach's $\alpha$ as well as the mean values and standard deviations for each item of the questionnaire. In the next step, we carried out ANOVAs in order to investigate the effects of gender and cultural background on self-concept. In the first analysis, we compared students with any migration background to students without a migration background. In the second step, we limited the analysis to the two largest cultural groups: students with a Turkish background and students without a migration background. We conducted an ANOVA, investigating the effects of gender and cultural background. All analyses were carried out using R version 3.4.2 [31], in particular, the packages car [32] and psych [33].

*Qualitative analysis.* The interviews were analyzed using qualitative content analysis [34] with MAXQDA software. We opted for a content-based approach and coded each part of speech that contained a new content. These units were paraphrased. In this step, we focused on goal orientations, feelings, and evaluations of abilities in chemistry. Units with similar content were omitted. In the third step, we organized the content in four categories: (1) learning goal orientations, i.e., thinking patterns that show the desire to engage in learning processes; (2) performance goal orientations; (3) social goals, i.e., thoughts about the social relations in class, and (4) the students' evaluations of their abilities (self-concept). Categories 1, 2, and 4 were defined prior to the analysis. Category 3 was defined during the analysis because it appeared to be a relevant topic in the interviews. In the fourth and last step, we checked the appropriateness of the categories, going through the data once again.

## 5. Results

### 5.1. Questionnaire Data

*Impact of migration background and gender on chemistry self-concept.* We analyzed the effects of gender and migration background on self-concept. For this analysis, we used the complete sample. Reliability was tested using Cronbach's $\alpha$ which was with 0.86 very good. Levene's test was not significant at the 0.05 level for the main and interaction effects. The Q-Q plots indicated a normal distribution of the data and the Shapiro–Wilk tests were not significant at the 0.05 level. We therefore conducted an ANOVA with the four groups of boys ($N = 25$) and girls ($N = 28$) without a migration background, and boys ($N = 28$) and girls ($N = 35$) with a migration background (Table 3). The main effect of gender, $F(1, 112) = 4.09$, $p = 0.046^*$, and the interaction effect were significant, $F(1, 112) = 6.38$, $p = 0.013^*$. The effect of culture was not significant, $F(1, 112) = 1.60$, $p = 0.209$. This first analysis shows that boys have significantly stronger self-concepts than girls, *M(male)* = 3.71 > *M(female)* = 3.48. Also, the relationships between the gender groups seem not to be the same among students with and without a migration background: in the subsample of students without a migration background, boys scored higher than girls (*M(boys)* = 3.98 > *M(female)* = 3.40). In the subsample of students with a migration background, however, there seems to be little difference between the gender groups (*M(male)* = 3.50 > *M(female)* = 3.54).

**Table 3.** Results of the two-way ANOVAs comparing the self-concepts of girls and boys with different cultural backgrounds. (* = p < 0.05)

|  | *F* | *p* |
|---|---|---|
| Analysis 1. Boys and girls *without* migration background vs. *with* migration background | | |
| effect of gender | 4.09 (1, 112) | 0.046 * |
| effect of migration background | 1.60 (1, 112) | 0.209 |
| interaction effect | 6.38 (1, 112) | 0.013 * |
| Analysis 2. boys and girls *without* migration background vs. *with Turkish* migration background | | |
| effect of gender | 4.59 (1, 75) | 0.035 * |
| effect of migration background | 2.39 (1, 75) | 0.127 |
| interaction effect | 5.04 (1, 75) | 0.028 * |

*Impact of Turkish migration background and gender on chemistry self-concept.* In the next step, we analyzed the effects of gender and having a Turkish background on chemistry self-concept. For this analysis, we used the subsamples of students without a migration background and with a Turkish migration background. Levene's test for the main and interaction effects was not significant at the 0.05 level for the main and interaction effects. The Q-Q plots indicated a normal distribution of the data and the Shapiro–Wilk tests were not significant at the 0.05 level. Just as in the previous ANOVA, the main effect of gender, $F(1, 75) = 4.59$, $p = 0.035^*$, and the interaction effect of gender and culture, $F(1,$

75) = 5.04, *p* = 0.028*, were significant (Table 3). The effect of culture was not significant, $F(1, 75) = 2.39$, *p* = 0.127. This suggests that among students with a Turkish background, gender relations might be inversed. The boys with a Turkish background scored lower (*M* = 3.32) than the girls with a Turkish background (*M* = 3.50) while among the students without migration background, the boys scored higher than the girls (*M(boys)* = 3.98 > *M(female)* = 3.40). However, it is important to keep in mind that sample sizes for students with a Turkish background were small (13 boys, 14 girls). The discovered effects can, therefore, show tendencies in these groups but require further investigation.

*5.2. Interview Data*

With the interview data, we wanted to explore the dynamics of chemistry self-concept. More precisely, we wanted to investigate the relation of self-concept to learning and performance goal orientations (LGO and PGO) in chemistry class. We focused on the students without a migration background and those with a Turkish background because these were the largest cultural groups in the analysis. Table 4 provides an overview of the results of the analysis, showing patterns in the categories (1)–(4).

In the analyses, we link the students' self-concept scores from the questionnaires to the interview data. The overall chemistry self-concept mean value was *M* = 2.41. Scores higher than the mean indicate a positive self-concept. If the score is lower than 2.41, the student has a more negative chemistry self-concept than average. The scores are provided in brackets (SC = … ).

5.2.1. Boys without Migration Background

*(1) Learning goal orientation* tended to be high (Table 4). The boys showed interest in chemistry content: "I find it interesting how substances combine to form a new substance, or how they change due to certain reactions" (interview 9b, SC = 3.56). Another student showed that he is proud of himself if he improves his skills in chemistry which is also a sign of an LGO: "[I am proud] when I manage to do something I haven't done before" (interview 10b, SC = 3.44).

**Table 4.** Overview of the results of the interview analysis.

| | No Migration Background | | Turkish Migration Background | |
|---|---|---|---|---|
| | **Male** | **Female** | **Male** | **Female** |
| *N* | 7 | 12 | 5 | 4 |
| (1) Learning goal orientation + strong … weak – | + | – | – | + |
| (2) Performance goal orientation + strong … weak – | – (exception: obtrusive thoughts in one case) | + | + | – (exception: in one case low persistence) |
| (3) Social context + relevant … irrelevant – | – | + | + | + |
| (4) Evaluation of chemistry abilities + positive … negative – | + | – | – | +/– |

*(2) Performance goal orientation* seemed to be comparably unimportant (Table 4). Only in one case did obtrusive thoughts appear. These can be negative side effects of performance goal orientations [11]. The student explained:

Once you have a problem you can't solve ( … ) you get stressed in that situation because then these thoughts come: what if I skip that one and keep on doing the other stuff? Will I later be able to do the one I've skipped? ( … ) that keeps stressing me (...). Then you got a bit of trouble understanding just anything. (interview 22a, SC = 3.22)

Despite these obtrusive thoughts, the student shows an above average chemistry self-concept (3.22). The thoughts, therefore, seem to have little impact on his overall belief in his abilities in chemistry.

*(3) The social dimension* of chemistry learning seemed to be of little relevance (Table 4). Social issues were almost not mentioned in the interviews when the students were asked to talk about their experiences in chemistry class. In contrast to all other groups, none of the boys without a migration background expressed social insecurity.

*(4) The evaluation of abilities* tended to be positive (Table 4). This was interpreted as a strong chemistry self-concept in the sense of the Shavelson model. One boy even expressed strong science aspirations: "I would like to try studying chemistry in college, to try at least, because it's interesting to me" (interview 21a, SC = 3.44). His self-concept score of 3.44 was far above the average. For this student, a strong chemistry self-concept seemed to be closely intertwined with chemistry-related career aspirations.

### 5.2.2. Girls without Migration Background

*(1) Learning goal orientation* tended to be low (Table 4). This showed, for instance, in a preference for easy tasks. One girl said that she feels "relieved" when she can do an easy task (interview 3a, SC = 1.89). Another girl seemed more learning goal oriented, but only slightly: she chooses to do easy tasks first "because that motivates me and then I do the difficult ones" (interview 2a, SC = 2.56). She needs a high motivation to work on difficult tasks which is a sign of rather weak learning goal orientation.

*(2) Performance goal orientation* seemed to be present (Table 4). This showed primarily through negative side-effects [35]. For instance, one girl showed an entity theory of intelligence [7], saying "I just feel stupid sometimes" (interview 5a, SC = 1.78). She did not talk about the things she needed to learn. Instead, her self-description as 'stupid' suggests that she sees herself as incapable of learning, which is an entity theory of intelligence.

*(3) The social context* seemed to be highly relevant (Table 4). This was in some cases coupled with ambiguous and rather negative feelings:

> I would say that [in contrast to other classes] there is maybe a bit more tolerance. There's no need to be so frightened to ask something that ( . . . ) you should know already. (interview 3b, SC = 1.78)

On one hand, this girl feels a lot of tolerance when someone makes a mistake. However, on the other hand, she feels that there are things she "should know already" which could reflect a feeling of pressure and insecurity. Although she thinks that there is no need to be frightened, this implies that she does not take it for granted to be accepted because she considers the possibility that she could feel frightened. Another girl showed a similar ambiguity. She said, "you just feel like you're one of them" (interview 3a, SC = 1.89). On the one hand, this girl wanted to express that she feels accepted in class. On the other hand, the fear of not being accepted appears to be very present to her and only banned for the moment. Another girl expressed her feeling of being an outsider in a drastic way: "you just feel as if you had no business there and that you're in the wrong place" (interview 5a, SC = 1.78).

*(4) The evaluation of abilities* tended to be negative (Table 4). Some girls said they had serious difficulties dealing with chemistry content.

> All these calculations and stuff, that's not my thing. I find it interesting but all this calculating with the ions switching from one side to the other, that's quite a bit confusing. (interview 3b, SC = 1.78)

Furthermore, the girls' self-concepts about their verbal and mathematical skills in chemistry seemed to be very mixed.

### 5.2.3. Boys with Turkish Migration Background

*(1) Learning goal orientation* seemed to be of little relevance (Table 4). Reflections on the learning process appeared very rarely in the interviews. One student liked easy tasks because in these situations, he "could help others, too" (interview 22b, SC = 1.67). This shows that content learning was not a high priority for this student.

*(2) Performance goal orientation* seemed to be rather strong (Table 4). Test performance seemed to be of high importance, as well as opportunities to show their abilities to the other students (interview 2b, SC = NA).

*(3) The social context* seemed to be relevant (Table 4). In one case, this was coupled with very difficult feelings. One of the boys (interview 22b, SC = 1.67) seemed to feel left behind. During the interview, he spoke a lot about feelings of uncertainty, nervousness, and being left alone. He described a situation when he prepared for a presentation with a group of students. However, his classmates abandoned the project, something he interpreted as "they let me down." This put him into a difficult situation because he is "rather nervous" when standing in front of the class. In such situations, the student tells himself that "nobody thinks bad of me" in order to "calm myself down". This reflects the student's preoccupation with his social position in the class. However, the positive side of this strong social focus showed in the fact that the student prefers working in groups.

*(4) The evaluation of abilities* was rather negative (Table 4). One of the students said, "we [him and the other interview partner who had a Turkish background as well] are not the smartest students in chemistry" (interview 6b, SC = 2.56). This tendency to describe the abilities as rather low was present in other interviews as well. One student said: "it [chemistry] is a bit difficult for me" (interview 22b, SC = 1.67), and it was the calculations in particular that put him to his limits. It is difficult for him to follow in class because he understands the content "only, like, partly" and sometimes gets really frustrated, angry, and sad about this.

### 5.2.4. Girls with Turkish Background

*(1) Learning goal orientation* seemed to be strong (Table 4). One student seemed to be engaged in deep learning processes in chemistry. She said, when "you get this mental representation, I like that part best." (interview 1b, SC = 2.56). She enjoys developing new ideas and representations of abstract entities which is a sign of a need for cognition. In addition, she likes understanding the theoretical background of experiments: "we see a change in the solution ( . . . ) and then we can maybe explain with a formula what happens" (interview 1b, 2.56). Another student expressed her desire to understand chemistry content this way: "well, I don't understand it *yet*. At some point, I'm gonna get it. It can't be that difficult" (interview 24a, SC = 3.11). Here, she shows an incremental theory of intelligence. She believes that she can improve her skills in chemistry.

*(2) Performance goal orientation* seemed to be of much less relevance compared to learning goal orientations (Table 4). Only one girl showed low task-persistence: "when I don't get it right from the start, then I try to see if . . . maybe I will understand it later, but when I, when I see I don't get it ( . . . ) I don't feel like making any more effort and it stays like that until the end of the lesson" (interview 1a, SC = 2.23). This can be interpreted as a sign of a performance goal orientation.

*(3) The social context* seemed to be very relevant (Table 4). For one student, her relationship with the teacher was important: "the classes are fun but I think it's more about the teacher than about the lessons as such" (interview 1a, SC = 2.23). She described her teacher as being very attentive and appreciated the teacher's efforts to actively work on the relationships with the students and to motivate them. Also, the two girls in this interview (1a, SC = 2.23 and 1b, SC = 2.56) referred to people outside school as being relevant for chemistry learning. They enjoyed sharing things they had worked on in class. For instance, two girls had produced a video during class about an experiment which they shared with people in their home environment. Chemistry learning, therefore, seemed to be easy to integrate into their social life.

*(4) The evaluation of abilities* was mixed (Table 4). Some students expressed a skeptical view on their abilities in chemistry, saying for example "I am not the best student in chemistry" (interview 23, SC = 2.44). One of these girls talked about her trouble to concentrate on the task when things get difficult. Others were rather positive.

## 6. Discussion and Conclusions

In this section, we will first discuss the two empirical research questions (EQ1 and EQ2) by summarizing the results of the study and interpreting them. Then, we will cover the theoretical question (TQ), discussing to what extent the approach can increase culture-sensitivity in academic self-concept research. In the last part, we will reflect on the study's limitations.

### 6.1. EQ1 and EQ2: Discussion of the Findings

In the analysis of the questionnaire data, we made three findings regarding EQ1 (How strong are the chemistry self-concepts of students with different cultural backgrounds and gender living in Germany?):

1.  Gender had a significant impact on chemistry self-concept. We replicated the gender gap that had been reported on students in Germany [15]. The gender gap in favor of boys that exists in science self-concept in most regions has inspired a number of intervention studies. However, progress in this area still seems to be limited [1]. Perhaps different types of intervention studies could be interesting, focusing on aspects that are closely related to self-concept but that might be easier to change. For example, it has been shown that learning goal orientations are closely related to self-concepts [36]. It could be interesting to focus on these orientations in interventions because they influence task-choice and learning behavior directly.
2.  The cultural background did not have a significant effect on chemistry self-concept. This is not in line with the finding that in most cases, the science self-concept of students who belong to a non-dominant cultural group is lower on average than that of students who belong to the dominant cultural group (e.g., [14]).
3.  The interaction effect of gender and culture on chemistry self-concept was significant.

The analysis suggests that the students' cultural background has an impact on chemistry self-concept in interaction with gender. More precisely, gender relations seem to differ between the groups of students without a migration background and those with a Turkish migration background. In the sample of students without a migration background, we found the gender gap that had been reported on, with boys having stronger self-concepts [15]. However, in the subsample of students with a Turkish background, the girls tended to show stronger beliefs in their abilities in science: their mean self-concept score was higher than the boys'. This finding had not been documented in the literature and was, therefore, unexpected.

What could explain that gender relations seem to be inversed in the sample of the students with a Turkish background? We looked into the interview data for phenomena that could help to develop hypotheses, in order to answer EQ2 (In what ways are the students' chemistry self-concepts associated with learning and performance goal orientations?). In the four groups, it seemed that different patterns were active:

*Pattern I: strong self-concept coupled with learning goal orientation.* This appeared in the group of boys without migration background who seemed to be highly learning- and content-oriented. In comparison, the social context appeared almost irrelevant to them. This pattern was associated with high self-concept scores in the questionnaire.

*Pattern II: strong self-concept coupled with learning and social goal orientation.* This appeared in the group of girls with a Turkish migration background. These girls had high self-concept scores in the questionnaires, too, but showed a different pattern: in addition to a strong orientation towards

chemistry content and learning, they had a strong social orientation. From their perspective, chemistry learning appeared as embedded in the social context.

*Pattern III: weak self-concept with strong social focus and feelings of insecurity.* This pattern appeared in the group of girls without migration background and boys with a Turkish background. It is characterized by concerns about social issues. This pattern was associated with low self-concept scores in the questionnaire and negative statements about their abilities in the interviews. This uncertainty about their abilities coupled with their social goal orientation leads many students to raise the question of social acceptance.

Knowing that chemistry self-concept can be associated differently with learning goal orientations and the feeling of social support could open new paths for application-focused self-concept research. It could be interesting regarding students who score very low on self-concept scales to explore the effects of cooperative learning and similar methods that try to enhance peer support. Also, task-choice behavior and other practical implications of learning goal orientations could be focused upon when trying to support the students' development of positive self-concepts. Insofar, the results of mixed methods design study could inspire more application-focused self-concept research which seems to be lacking [1].

### 6.2. TQ: The Culture-Sensitivity of the Approach

To what extent can the concurrent triangulation mixed methods approach provide a basis for culture-sensitive academic self-concept research? (TQ). In the present study, the approach helped to uncover a possible conceptual non-equivalence [2] in chemistry self-concept. Chemistry self-concept seemed to be related in different ways to (i) gender and to (ii) the salience of the social structure in chemistry class. In the subsample of students without a migration background, strong self-concepts seem to be possible only if the social structure is ignored. Speaking in stereotypes, one could conclude that in this subgroup, positive self-concepts are only possible for boys if they learn to focus on chemistry content while not considering the social structure. In contrast, in the Turkish subsample, gender seemed to be associated differently with chemistry self-concept. Chemistry appeared as slightly more accessible for the girls because they showed quite positive self-concepts and strong learning goal orientations. In addition, dissociating chemistry learning from the personal social context seemed not to be necessary for having a positive chemistry self-concept. The girls with a Turkish migration background who had high self-concept scores tended to show learning goal orientations with a focus on chemistry content and, at the same time, a strong focus on the social domain. It seemed to be possible for these girls to develop a personal connection with chemistry and to enjoy it as part of social life. Our hypothesis is, therefore, that chemistry might be perceived as more gender-neutral among the students with a Turkish migration background. In addition, it seemed to be perceived more strongly as part of social life. This shows that conceptual equivalence might not be given when analyzing chemistry self-concepts in these two cultural groups.

It is striking that the students' cultural background seems to play such an important role in their attitude towards chemistry. All students in our sample went to schools in the same country. They therefore experience chemistry in the same educational system. Most of the students with a Turkish migration background were born and raised in Germany and only their mother and/or father were born in Turkey.

We considered further literature in order to build a hypothesis of what might produce these differences. Our search made us aware of the fact that in Turkey, slightly more young women than men hold science degrees [37]. In addition, the girls achieved substantially better in science than the boys in the PISA 2015 [38] and TIMSS 2015 study [39]. This suggests that science might be perceived as more gender-neutral in Turkish society. Our hypothesis is that this different association of science with gender might influence the chemistry self-concepts of students with a Turkish background. This underlines the need for culture-sensitive science self-concept research.

Measurement non-equivalence seemed not to be present. Firstly, the students' oral descriptions of their abilities in chemistry seemed to correspond to their self-concept scores. The self-concept scores from the questionnaire, therefore, seem to be a good gauge for students' self-concepts in both cultural groups. Secondly, thinking about self-concept seemed to be equally salient in both cultural groups. This suggests that self-concept is of equal importance in both cultural groups. Measurement bias seems to be of minor importance in this study. This could possibly be due to the fact that all students live in Germany and most of them have probably been raised in the country. The students are, therefore, presumably equally familiar with Likert-type items.

The concurrent triangulation mixed methods design proposed in this study has several advantages compared to self-concept research that relies on Likert scale data only. (i) It can uncover conceptual non-equivalence as it did in this exploratory study; (ii) It can uncover measurement non-equivalence which seemed not to be present in our study; (iii) It pushes researchers to ask questions that go beyond the description of differences in self-concept. This can lead to more culturally embedded analyses. The present investigation constitutes an example of the attempt to embed self-concept data into the cultural context through trying to understand the culturally different associations of the chemistry with masculinity.

*6.3. Limitations of the Study*

The present study was exploratory in nature with the goal to assess the usefulness of a concurrent triangulation mixed methods design in self-concept research. Due to its exemplary nature, it faces several difficulties. One limitation of the study is the sample size, which poses difficulties, especially to the statistical analyses. However, the fact that we replicated the overall gender gap in favor of the boys encouraged us in our further analyses. It had been documented numerous times for Germany and, therefore, suggested that our sample represents some key trends. More importantly, the interviews support the assumption that the quantitative data are of good quality: many students talked about their abilities in the interviews. Their verbal statements corresponded to their self-concept scores in the questionnaire. Students who in the interviews talked about confidence in their abilities in chemistry showed self-concept scores above the mean. Students who were skeptical about their abilities in the interviews showed self-concept scores below the mean. This triangulation confirms the quality of the data. However, in order to be sure about the existence of the interaction effect between gender and cultural background, further investigations would be needed since this is the first time this has been documented.

Further, it is important to notice that the statement about the association of chemistry with masculinity that might differ between the two cultural groups is a hypothesis. At present, we cannot claim its validity because the interview study was explorative in nature. In order to claim validity for the hypothesis, a study aimed at hypothesis testing would be needed. The explorative approach was chosen because the primary goal of this study was to test a newly developed methodological and theoretical framework for more culture-sensitive academic self-concept research.

One strength of the approach is that it provides a framework for self-concept investigations in a mixed methods design with considerable explanatory power. This is due to the fact that the qualitative data are not used as illustrations but make substantive contributions to the whole analysis. In this study, the interviews served two purposes: (i) they allowed assessing the accuracy of the results from the quantitative part of the study. We saw that the students' statements about their abilities in the interviews corresponded to those in the questionnaire. In addition, it made us aware of conceptual non-equivalence. This is a triangulation assuring the validity of the data; (ii) The interview data allowed us to explore possible meanings of the findings from the questionnaire data. As stated above, in this study, we could derive hypotheses. However, we cannot claim their validity because this is the first time this approach was employed in a study. Despite this, the analysis of the interviews according to Hattie's process-oriented framework resulted in important insights. It made us aware of differences in gender conceptions that might play a role. We, therefore, believe that the approach to

academic self-concept research explored in this article could lead to more culture-sensitive research and a deeper understanding of academic self-concept. Of course, the adoption of other models could be good alternatives, such as identity models which integrate self-concept in their conception. This could be fruitful because identity theories tend to focus on the processes of identity construction and sociocultural narratives whereas self-concept research provides quantitative information. Exploring the scope of joint approaches could advance self-concept research. We suggest that the presented approach could be used in other domains in order to further explore its explanatory power.

**Author Contributions:** Conceptualization, L.R.; Formal analysis, L.R.; Investigation, L.R.; Methodology, L.R.; Supervision, S.M.; Visualization, L.R.; Writing – original draft, L.R.; Writing – review & editing, L.R. and S.M.

**Funding:** This research received no external funding.

**Acknowledgments:** We would like to thank all students and their teachers who participated in this study.

**Conflicts of Interest:** The authors declare no conflict of interest.

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
