# Peer review of "A Mixed Methods Approach to Culture-Sensitive Academic Self-Concept Research"

_education, doi:10.3390/educsci9030240_

Round 1
Reviewer 1 Report
The article may contribute a lot to the journal's readers. The methods are good, and the rationale is fine, although the ideas are not innovative. However, there is already quite a large body of knowledge related to this subject. Therefore, I recommend expending and up-dating the literature review.
Author Response
„The article may contribute a lot to the journal's readers. The methods are good, and the rationale is fine, although the ideas are not innovative. However, there is already quite a large body of knowledge related to this subject. Therefore, I recommend expending and up-dating the literature review.“
Dear reviewer,
Thank you very much for reviewing our manuscript. It is true that a large body of literature on science self-concepts exists. We have discussed the literature in depth in a literature review that has recently been published. We now refer to this in lines 25 and 54/5. Also, regarding the theoretical foundations of self-concept and learning goal orientations, we discuss only the fundamental literature (Shavelson et al. and Dweck et al.) since these authors still constitute the theoretical basis for research in this area despite the fact that their major works stem from the 1970s and 80s. We clarify this in lines 66/67, 72, and 99-101. We hope to have improved the manuscript according to your suggestions.
Kind regards,
The auttors
Reviewer 2 Report
Overall, this article is good. The purpose of the article is very interesting and relevant to the advancement of knowledge and research about chemistry self- concept. Theoretical and diversified references are correctly used. However, just some minor adjustments are requested.
Dear Author/Editor,
Thank you for your time and effort. There are several concerns that I have within the present manuscript: Please assist us by clarifying the following queries:
OBSERVATION 1.
GENERAL
The title, abstract, …. and general manuscript seems to be focused on culture-sensitive academic self-concept research. However, in your research i think the goal is to explore the approach for a more culture-sensitive academic chemistry self-concept. Thus, it is better to focus chemistry self-concept than general self-concept in your paper.
OBSERVATION 2.
1. Introduction. Theoretical Background
Why do the authors refer to methodological difficulties in self-concept research? Were the authors design and validate a scale or questionnaire?
I think they intend to justify the type of design. However, I think they should do it later. So please reconsider omit or change this information in the manuscript.
OBSERVATION 3.
Research goal and questions. About figure 1: Selected hypotheses regarding possible influences of culture on science self-concept. Are your hypotheses? It is a theoretical model proposed?
I think that the goal and questions should be summarized and clarified better in your study.
4. Materials and Methods
4.2. Instruments. Questionnaries.
Please include information about the reliability. What was the cronbach’s alpha reliability coefficient in this study? Please also provide information on ranging, name of the scale, authors, reference.
4.3. Sample
The authors provide a very detailed account of how participants were recruited and sampled. However, please also provide the reference of the Declaration of Helsinki (line 235). I think it is (Medical, A.W. Declaration of Helsinki Ethical Principles for Medical Research Involving Human Subjects.J. Am. Med. Assoc. 2013, 310, 2191–2194).
Please, it would be convenient to inform about the type of school (public, private or concerted). In addition, it should provide information on the mean and standard deviation of the age of the girls and boys in the sample.
4.4. Data Analysis
It would be interesting to know the version of the program (R). Also, it would be interesting which program you have used for the Qualitative Content Analysis
OBSERVATION 4.
5. Results
The authors provide information about their results were supported. However, there was quite a bit confusing in the “ANOVA section”– please work to interpret your results and incorporate an ANOVA table in this part.
I suggest one of my reference authors for this work and for the next, Fernando Garcia, Ph.D., is Full Professor of Psychological Methods and Design of Research. Orcid = https://orcid.org/0000-0002-3953-8364.
Also, I provide an example of how i would do it. For example:
As shown in Table X, the boys obtained higher average scores than girls in chemistry self-concepts F(1, 112) = 4.09, p < 0.005, ……No significant effects was observed for culture.
Table X. Example table.
|
Variables |
Sex |
p |
F(1, 112) |
η2 |
|
|
Male |
Female |
||||
|
Without a migration background |
M (SD) |
M (SD) |
Provide |
Provide |
Provide |
|
A migration background |
M (SD) |
M (SD) |
Provide |
Provide |
Provide |
*** p < 0.001; ** p <0.01; * p < 0.05
OBSERVATION 5.
5. Discussion and conclusion
I believe that this result offers progress in an area that is still fairly under explored and of particular importance in terms of intervention from schools. Particularly, the results the pattern III “weak self-concept with strong social focus and feelings of insecurity” are really interesting. But, i think that these results and these theoretical and practical implications were not enough explained or used in your study. However, the introduction is good. There is a disconnection between the introduction and the discussion and conclusion.
Although gender was not analyzed importantly in this study, the result was on gender difference was significant. The discussion is very poor and the practical implications very powerful. Please expand this aspect.
I suggest you modify this phrase (line 522 and 523) “ IT IS SURPRISING THAT CULTURE SEEMS to have such an impact on the students. This underlines the need for culture-sensitive science self-concept research". There are many serious works that have found that the relationship with culture and this result. The reader might think that you have not performed a serious search.
Finally, (line 522 and 523), “It pushes researchers to ask questions that go beyond the description of differences in self-concept. This can lead to more culturally embedded analyses”…For example?
I think that the article does not include a good discussions about the practical and theoretical implications of the results. Please, the work is good and powerful. Consider my suggestions!
Author Response
Dear reviewer,
Thank you very much for this in-depth analysis of our article and your comments which helped us improved the manuscript. We have tried to respond to all of your observations and have made several amendments to the manuscript. We hope this meets your expectations.
Kind regards,
The authors
Observation 1
“The title, abstract, …. and general manuscript seems to be focused on culture-sensitive academic self-concept research. However, in your research i think the goal is to explore the approach for a more culture-sensitive academic chemistry self-concept. Thus, it is better to focus chemistry self-concept than general self-concept in your paper.”
It is true that the empirical study concerns chemistry self-concept. However, the reason for preparing this manuscript for Education Scienceswas that we propose a new approach for culture-sensitive academic self-concept research irrespective of the concrete subject that is studied. It can be applied to academic self-concept research in all subjects. We have pointed this out in line 35-37.
Observation 2
“Why do the authors refer to methodological difficulties in self-concept research? Were the authors design and validate a scale or questionnaire?
I think they intend to justify the type of design. However, I think they should do it later. So please reconsider omit or change this information in the manuscript.”
In the present manuscript, we do not intend to present a new research instrument. However, we discovered severe methodological difficulties in a review of current self-concept literature (Authors, 2019) that we refer to in line 25. In the paper we propose one possible solution to this problem. We propose a different approach to academic self-concept research based on a mixed methods design. This is a central aspect of the present manuscript since it constitutes the innovation we introduce to the existing literature. This is the reason why we introduce the idea at the very beginning of the manuscript. We tried to clarify this in lines 30/31 and 33-35.
Observation 3
Figure 1: We tried to clarify the origin of the hypotheses in the caption of figure 1. “I think that the goal and questions should be summarized and clarified better in your study.” We tried to do this in lines 190/191 and 195-197. Questionnaires: More information on the used scale is provided in lines 219 and 222/223. Sample: We included the reference to the Declaration of Helsinki in line 247. We included more information on the school types and ages in line 259. Software We included the R version and the software with which qualitative data analysis was conducted in lines 293 and 295/6.
Observation 4
Thank you for providing the ANOVA table, it gives a good overview of the results. We included an ANOVA table conforming to the APA standards on page 8.
Observation 5
“the results the pattern III “weak self-concept with strong social focus and feelings of insecurity” are really interesting. But, i think that these results and these theoretical and practical implications were not enough explained or used in your study.” We provide a discussion in lines 515-522. “Although gender was not analyzed importantly in this study, the result was on gender difference was significant. The discussion is very poor and the practical implications very powerful. Please expand this aspect.” We included the discussion of this aspect in lines 471-478. I suggest you modify this phrase (line 522 and 523) “IT IS SURPRISING THAT CULTURE SEEMS to have such an impact on the students. This underlines the need for culture-sensitive science self-concept research".” Thank you for this comment. The phrase was indeed misleading and we have reworked the text accordingly. “Finally, (line 522 and 523), “It pushes researchers to ask questions that go beyond the description of differences in self-concept. This can lead to more culturally embedded analyses”…For example?” We tried to clarify this aspect in lines 568-570.Reviewer 3 Report
First of all I would like to congratulate the authors of the article for the excellent work done. The manuscript presents an important number of strengths that I would like to highlight.
The study is inserted in an important and current line of research on the “academic self-concept”. The study uses a mixed methods design (questionnaire and interviews) with the aim of examining the effect of different cultures (validity). The findings found are very interesting. The data provided are very useful for practical intervention, especially for gender differences (gender gap in favor of boys).
All sections of the manuscript (introduction, methods, results, discussion, conclusions, references) have been prepared successfully. The data analysis is correct and is consistent with the hypothesis.
Congratulations for the work done.
Author Response
“First of all I would like to congratulate the authors of the article for the excellent work done. The manuscript presents an important number of strengths that I would like to highlight.
The study is inserted in an important and current line of research on the “academic self-concept”. The study uses a mixed methods design (questionnaire and interviews) with the aim of examining the effect of different cultures (validity). The findings found are very interesting. The data provided are very useful for practical intervention, especially for gender differences (gender gap in favor of boys).
All sections of the manuscript (introduction, methods, results, discussion, conclusions, references) have been prepared successfully. The data analysis is correct and is consistent with the hypothesis.
Congratulations for the work done.”
Dear reviewer,
Thank you for your message and your kind words. Since the other reviewers discussed certain aspects of the article, we made several changes to the original manuscript. We hope it still meets your expectations.
Kind regards,
The authors